# Imitation recognition and its prosocial effects in 6-month old infants

**Gabriela-Alina Sauciuc**[1]*, **Jagoda Zlakowska**[2], **Tomas Persson**[1], **Sara Lenninger**[3], **Elainie Alenkaer Madsen**[1]

**1** Department of Philosophy, Cognitive Science, Lund University, Lund, Sweden, **2** Faculty of Humanities, Nicolaus Copernicus University, Torun, Poland, **3** Department of Educational Sciences, Kristianstad University, Kristianstad, Sweden

* Gabriela-Alina.Sauciuc@lucs.lu.se

**Data Availability Statement:** All relevant data are within the manuscript and its Supporting Information files.

**Funding:** This study was funded by the Swedish Research Council (Vetenskapsrådet www.vr.se),

## Abstract

The experience of being imitated is theorised to be a driving force of infant social cognition, yet evidence on the emergence of imitation recognition and the effects of imitation in early infancy is disproportionately scarce. To address this lack of empirical evidence, in a within-subjects study we compared the responses of 6-month old infants when exposed to ipsilateral imitation as opposed to non-imitative contingent responding. To examine mediating mechanisms of imitation recognition, infants were also exposed to contralateral imitation and bodily imitation with suppressed emotional mimicry. We found that testing behaviours—the hallmark of high-level imitation recognition—occurred at significantly higher rates in each of the imitation conditions compared to the contingent responding condition. Moreover, when being imitated, infants showed higher levels of attention, smiling and approach behaviours compared to the contingent responding condition. The suppression of emotional mimicry moderated these results, leading to a decrease in all social responsiveness measures. The results show that imitation engenders prosocial effects in 6-month old infants and that infants at this age reliably show evidence of implicit and high-level imitation recognition. In turn, the latter can be indicative of infants' sensitivity to others' intentions directed toward them.

## Introduction

Imitation is frequently experienced by infants in their daily interactions with adults during the first year of life [1,2]. Given its pervasive presence in early infancy, the experience of being imitated is commonly regarded as a major driving force of infant social cognition. It has been proposed that constant exposure to being imitated by caregivers provides scaffolding for several socio-cognitive competences, including self-awareness, self-other discrimination [3–5], intentional imitation [1,2,6–8], the understanding of others' intentions [4,5] and the acquisition of turn-taking structure and of cultural norms [1,2]. Theories hold that the experience of being imitated enables infants to discover that some of their actions have external consequences, thus strengthening their sense of causality and agency [3–5]. Moreover, it has been argued

grant no. 2012–1387 secured by EAM, GAS, TP and SL. The funders had no role in study design, data collection and analysis, decision to publish, or preparation of the manuscript.

**Competing interests:** The authors have declared that no competing interests exist.

that, when experiencing self-produced actions along with nearly identical actions produced by caregivers, infants are provided with opportunities to connect their proprioceptive experience with the visually perceived responses of the caregiver, thus coming to an understanding of what their own actions look like [4,5,7]. The recognition of self-other matched behavioural states would, in turn, gradually facilitate an understanding of shared psychological states and intention reading [4,5]. Finally, it has been speculated that exposure to being imitated exerts a cultural shaping function [2]. As imitation by caregivers is selective, certain behaviours that are produced by the infant are repeatedly highlighted when imitated by others, thus potentially becoming familiar elements of interpersonal exchange.

As summarised above, the theorised developmental significance of imitation recognition is considerable, and has potentially far-reaching implications. However, with the exception of descriptive accounts and uncontrolled observations, there is a lack of compelling evidence that young infants reliably distinguish imitative interactions from other types of contingent responding, or that imitation at all has any effect on young infants. Although caregivers imitate infants frequently, these imitations are usually fleeting, and it is not until 8 months of age that prolonged imitative exchanges between caregivers and infants become common [1,2]. It is thus unclear if the brief imitations of early infancy are sufficiently salient to be readily detected by infants in the ongoing stream of caregiver behaviours. Extant research provides an incomplete picture of the phenomenon. Naturalistic studies on the presence of imitation in early mother-infant interaction do not include systematic analyses of the infants' reactions in response to imitation [1,2]. Even when more systematic analyses are conducted [9,10], only limited conclusions can be drawn from them, as these rely on uncontrolled observations. Based on this data, it is thus not possible to establish if the infants show imitation recognition or if they react to the temporal contingency of the caregiver's response, to the familiarity of the action or to channel congruence. The few studies in which imitation is used as an experimental condition provide contradictory results, reporting either heightened [11,12] or lowered [13] levels of attention when young (i.e. 3-month old) infants are imitated by their mother compared to spontaneous mother-infant interaction. Imitation is also found to elicit less smiling than spontaneous interaction [12,13], with one study finding this effect to hold only for highly attuned mother-infant dyads [13].

Systematic research on imitation recognition and the effects of being imitated is needed in order to tease apart the interactional ingredients that facilitate early adult-infant engagements and are likely to boost social cognition. Observational studies suggest that the experience of being imitated is initially highly affective in nature, with caregivers selectively and involuntarily mirroring infant facial and vocal behaviours that are perceived to have emotional content [1–3, 14,15]. Recent research has also shown that levels of received and produced facial mimicry co-vary in 4-month old infants [16], and that levels of exposure to facial mimicry at 2 months of age predicts neural processing of observed facial expressions at 9 months of age [17]. It is likely that this constitutes an ancestral behavioural mechanism of normal socio-cognitive development, as, for example, facial mimicry was found to improve levels of attention and affiliative responses in infant macaques raised in a socially impoverished nursery environment [18]. The effects of other channels and forms of imitation remain, however, understudied in early infancy, although observations suggest that, from about 6 months of age, infants are increasingly exposed to manual gesture imitation [1,2]. At this age, there is also a switch in infants' preferential focus of attention, from an initial preference for faces to an increasing focus on hands [19]. It is thus important to investigate infants' sensitivity to distinct forms of imitation, as a function of e.g. channel and preferred attentional focus.

Based on previously available data from studies with older infants, implicit imitation recognition is not reliably attested prior to 8 months of age [6]. At this age, infants have been found

to respond with higher levels of attention and smiling when imitated by their mother, compared to when the mother's response was temporally contingent upon the infant's action but non-imitative. From 9 months of age, infants have been found to show explicit imitation recognition, by exhibiting so-called 'testing behaviours' [4,5,20]. Testing behaviours comprise a class of responses by which the imitated individual is intentionally 'testing' the behavioural correspondence between him/herself and the imitator. For example, (s)he reproduces a previously imitated action several times (behavioural repetition), modifies it by e.g. speeding up or slowing down (behavioural modulation), or performs a rapid series of actions (testing sequence). Testing behaviours have thus been interpreted as an indication that the imitated individual is aware of the imitator's intention to copy his/her actions [4,5]. Younger infants (range 6 weeks—8 months) do not appear to exhibit such behaviours, at least not when they are simultaneously exposed to an imitating experimenter and a contingently responding one [5]. In fact, when this juxtaposed design is used, younger infants might not even show signs of implicit imitation recognition [5]. This runs against hypotheses proposed in the literature [7,10] that certain specific responses to being imitated (e.g. monitoring the imitator's actions and behavioural repetitions) would emerge already between 5 and 7 months of age. The simultaneous presentation of complex stimuli is, however, challenging to young infants. A sequential design could be thus more appropriate for tracking the presence of imitation recognition in younger infants.

In the present study, using a within-subjects design, we compared the responses of 6-month old infants when exposed to ipsilateral (i.e. Mirror) Imitation (MI) as opposed to (non-imitative) Contingent Responding (CR). These are the two key conditions used in previous studies of imitation recognition with infants [4,5,11,12,13,20]. The relevance of CR to studies of imitation recognition is grounded on several decades of research that have highlighted maternal CR as the key feature of mother-infant interaction that promotes infant socio-cognitive development [for reviews and discussions, see e.g. 21,22,23,24 and references therein]. This research shows that early mother-infant interactions exhibit a distinct temporal organisation, whereby infant behaviours are contingently responded to by the mother, most often with less than a second latency [21,22]. This temporal patterning of responses appears to shape the infant's communicative skills. As infants mature, they too begin to exhibit CR, and become sensitive to perturbations that affect the temporal organisation of interactions [e.g. 22 and references therein]. Data from research on e.g. *affect attunement* and/ or *maternal mirroring* suggests, however, that the *form* of maternal CR is just as crucial as its timing [22,23]. *Affect attunement* and *maternal mirroring* are composite constructs (and measures) that both refers to maternal behaviours which reflect a sharing of affect with the infant, either by mimicking the infants' affective expressions or by matching the emotional tone of those expressions, but in a different channel [3,13,14,23,24]. This aspect of maternal behaviour has been revealed to positively influence infant social responsiveness and sensitivity to the mother's social signals beyond the effects of generic CR [3,13,14, 23,24].

Imitation is yet another *form* of CR, which, as reviewed further above, has been theorised to be a driving force of infant socio-cognitive competence. In experimental settings, the key difference between imitation and CR is that in CR the infant controls the 'when' of social response, while in an imitative condition, the infant controls both the 'when' and the 'what' of the social response. It is currently unknown if 6-month old infants can discriminate between these forms of social interaction, and if these have different effects on infants' social responsiveness. Following references [7,10], we hypothesised that 6-month old infants will show *implicit* discrimination of being imitated, by showing a higher level of non-specific social behaviours (attention, smiling) in the MI condition as opposed to the CR condition. Since imitation promotes heightened levels of prosociality in older infants [25], we further sought to

investigate the developmental roots of this phenomenon. To this end, we set to measure approach behaviours throughout the experiment, considering that proximity to experimenter has been successfully employed as a measure of imitation-induced prosociality in studies with nonhuman primates [26]. Finally, following [7,10], we also hypothesised that 6-month old infants will evidence at least a rudimentary form of *explicit* imitation recognition, by exhibiting monitoring and behavioural repetition when being imitated.

To explore potential mediating mechanisms of imitation recognition, we added two novel conditions: Contralateral Imitation (CI) and Bodily Imitation with suppressed emotional mimicry (BI). The CI condition was aimed at controlling for spatial compatibility effects, i.e. that the infants did not discriminate imitation from contingent responding based on low-level features, such as the spatiotemporal correspondence between their own and the experimenter's actions, as opposed to recognising the structural similarity between those actions. Such effects have not been controlled for in previous studies on imitation recognition in children. Spatial compatibility, however, appears to critically mediate contingency detection, as infants (aged 3- and 5-months old) cease to discriminate contingent from noncontingent action displays when these are inverted on the left-right axis [27,28]. Research with adults further indicates that mirror imitation is perceptually prioritised, as—unlike contralateral imitation—it is recognised even when not explicitly attended to [29]. Moreover, it has been proposed that mirror imitation offers greater interpersonal facilitation [30], as prosociality effects have been found in adults who were surreptitiously exposed to ipsilateral mimicry, but not in adults exposed to contralateral mimicry [31].

In the BI condition, the infants were exposed to selective mirror imitation of bodily (hand, trunk) responses, while facial and vocal mimicry was suppressed. This provided infants with contradictory emotional signals concerning the experimenter's engagement, thus offering an additional way to assess if the infants' responses could be explained as low-level discrimination or high-level imitation recognition. The BI condition capitalises on several relevant transitions that take place around 6 months of age. Firstly, caregivers significantly increase their imitation of infant manual gestures, as opposed to earlier ages when infants' experience of being imitated is dominated by emotional mimicry [1–3,14,15]. Secondly, infants show diminished attention to others' faces, and increasingly attend to others' hands [19]. Finally, while it is generally acknowledged that facial and vocal signals routinely provide infants with affective commentaries on the ongoing (inter)actions that they experience [32], from 5 months of age, infants also exhibit recognition of bodily expressions of emotions [33]. Given these transitions, the BI condition presented an experimenter whose body was fully responsive and playful, while her face remained silent and immobile. If infants exhibited high-level imitation recognition, then the BI condition should selectively affect measures of social responsiveness (attention, smiling, approach), but not measures of explicit imitation recognition (testing behaviours). However, if the infants made an implicit discrimination, based on low-level stimulus features, then both implicit and explicit measures should be affected.

## Methods

### Participants

Sixteen infants (5 females, 11 males) aged 6,5-months (M = 199 days, SD = 3.425) participated in the study. Data from 12 additional infants were excluded from the study due to infant fussing (N = 2), equipment malfunction (N = 3), or third-party interference (by parent, sibling, mailman, etc.) that led to experiment interruptions (N = 7). The sample size was based on previous studies on imitation recognition that used a within-subject design [20]. A power analysis conducted in G*Power indicated that N = 16 was the minimal desired sample for a one-way

repeated-measures ANOVA, given $\alpha = 0.05$, a power of 0.8, and $\eta_p^2 = 2.1$. An additional analysis (computed following [34]) indicated that, for a Friedman's ANOVA, N = 16 was well above the minimal desired sample given $\alpha = 0.05$, a power of 0.80, and an effect size corresponding to the lowest limit of the medium range.

## Procedure

Infants were tested at home while seated on their mother's lap, at a table. The experimenter sat adjacently, at a 90-degree angle and approximately 50 cm from the infant. All sessions were video-recorded by three cameras, which focused on the infant's body, the infant's face and the whole scene, respectively. As compensation for their participation, at the end of testing, the family received a toy. Each infant visit comprised three phases: preparation, experimental phase and debriefing.

**Preparation.** The purpose of this phase (which lasted approximately 15 minutes) was to familiarise the infant with the experimenter, to provide the parent with instructions, and set up the cameras. The experimenter engaged the infant immediately upon arrival, by making eye-contact, smiling, and calling the infant's name. The experimenter then explained the experiment to the mother and instructed her to allow the infant to respond freely, regardless of whether the infant stood up, withdrew from the interaction, pulled the experimenter's hair, etc. The mother was also informed that the experiment would be stopped if the infant fussed longer than 30 seconds. While talking to the mother, the experimenter made eye contact with the infant as well, smiled and nodded toward the infant.

**Experimental phase.** The experiment used a within-subject design with four conditions presented in counterbalanced order, including: (i) Mirror (i.e. ipsilateral) Imitation (MI); (ii) Contralateral Imitation (CI); (iii) Bodily Imitation with suppressed emotional mimicry (BI); (iv) Contingent Responding (CR). In order to capture testing behaviours, each experimental condition lasted two minutes, since, in the only previous study that measured latency to the first testing bout, 18-month old infants initiated testing behaviours in response to being imitated with a mean latency of 68.9 seconds [35].

In the MI condition (S1 Video), the experimenter tried to ipsilaterally imitate all bodily actions, facial expressions and vocalisations produced by the infant. This condition provided the highest level of matching, as the experimenter matched infants' actions on a temporal (timing and rhythm of action), spatial (same axis), morphological (action shape), topographical (effector) and affective level. The CI condition (S2 Video) was similar to MI, with the exception that the infant's actions were imitated contralaterally. In the BI condition (S3 Video), the experimenter imitated ipsilaterally all infant's bodily actions, while all affective (facial and vocal) feedback was suppressed. In the CR condition (S4 Video), the experimenter responded whenever the infant acted, matching the temporal features of the infant's behaviour, but without copying the infant's response. The condition was standardised so that a predetermined inventory of three actions were presented during the condition, being rotated each 20 seconds. These actions included both elements that are common in mother-infant interaction [1] (waving while saying 'Hello' (i.e. *hej* in Swedish), and mouth-opening accompanied by a popping sound), as well as novel ones (face tapping while saying the nonce word *koll*). All three actions were accompanied by vocalisations in order to increase their salience and to mimic CR in mother-infant interaction, considering that vocalizing in response to the infant's actions is by far the most common type of maternal CR in such interactions [22]. To prevent the theoretical possibility that the form of the experimenter's CR matched, by chance, the form of an action produced by the infant, the experimenter was allowed to switch to a different action from the inventory should such a chance event occur. It must be pointed out, however, that the

likelihood of such a coincidence was extremely low. The actions that were selected for the study, while not necessarily novel to the infants, are not part of the typical motor repertoire of infants at the targeted age. Six-month old infants generally do not wave, nor do they produce mouth openings with a popping sound. They have, however, the capacity to learn such new gestures, but only after intensive modelling and training [36].

It is important to highlight that, in previous studies on infant imitation recognition, CR has received different operationalisations. Indeed, in the three studies with 3-month old infants [11, 12,13], CR was operationalised as mother-infant interaction, whereas in studies with older infants CR was operationalised as either minimal feedback [6], or as selective responding that is temporally contingent only upon certain behaviours produced by the infant but different from them [4,20] (see further down for more details).

The reasons for the former operationalisation are straightforward, as it follows a long research tradition which has emphasised that the efficacy of mother-infant interaction in shaping infant socio-cognitive competence is due to maternal CR [21,22]. However, given the within-subjects design of our study, in which the infants interacted with an experimenter, the former approach (CR as mother-infant interaction) was not applicable. This approach, moreover, presents the disadvantage that the effects of CR cannot be disentangled from familiarity effects. Indeed, the level and content of CR is highly variable across mother-infant dyads [22, 37], and it has been found that, during spontaneous interactions with strangers, infants show higher levels of social responsiveness when the stranger exhibits CR levels and contents that are similar to those to which they are accustomed [37]. Moreover, since during spontaneous interaction, besides CR, mothers produce also a variable amount of directive initiatives [38], it remains unclear whether it is the contingent feedback of the mothers or the mothers' attempts to structure the interaction that drives infants' social responses during such interactions. Consider, for example, how much attention and smiling scores could be affected by the frequent directive of calling the infant's name.

The uniform minimalistic feedback used in the study with 8-month old infants consisted in the phrase 'Yes, good!', which was consistently uttered by the mother every time the infant produced an action [6]. Since this feedback was not repeated several times when infants produced repetitive actions, it failed to capture the temporal properties of the infants' behaviours. In turn, it can be argued that this could have made it more difficult for the infants to detect the contingent relationship between their own and the mothers' behaviours, given the lack of temporal match between these behaviours whenever infants produced repetitive actions. In the selective responding case, which has been used in studies with 9-, 14- and 18-month old infants [4,20], the experimenter responded with one of several predetermined behaviours, but only to certain actions produced by the infant. In other words, whenever the infant did A, the experimenter responded with action X, whenever the infant did B, the experimenter responded with Y, and so forth. Thus this approach too presents the risk of dampening the salience of temporal contingency, as is likely to drive the infants to associate a specific response of their own with a specific response of the experimenter, rather than highlighting the temporal contingency relationship between the infants' actions and the experimenter responses. Finally–and similar to operationalisations of CR as spontaneous mother-infant interaction— this approach also limits the degree to which the infant controls the interaction. Thus the infants' higher levels of social responsiveness when being imitated (compared to the CR condition) could be explained instead by the higher levels of control over the interaction afforded by the imitation condition.

With our design, we tried to combine elements of social and temporal contingency that were present in previous studies, while trying to circumvent some of their limitations. We attempted to maximize the level of CR in order to allow the infants to more easily detect

temporal contingency, and to allow them to control the interaction to the same degree as they did in the imitation conditions. Just like in [4] and [20], we employed a predetermined inventory of behaviours. These, however, were not deployed selectively only if the infant performed specific actions, but were deployed to all actions performed by the infant, in a blocked fashion. We judged (and observed in pilots) that this blocked approach (i.e. rotating actions about 20-sec at the time) would give sufficient time to infants to notice the temporal contingency of the behaviours produced by the experimenter, while at the same time counteracting the unwanted effects of habituation and boredom. Indeed, every time a new action started, the infant's attention would be drawn back to the interaction.

The studies that operationalise CR as spontaneous mother-infant interaction [11,13,14] inspired us to use both actions that were likely to be familiar to the infants, as well as one action that was novel. Further following these studies, but also [6], as well as the literature on social contingency detection, we tried to emulate the most commonly observed elements of contingency in mother-infant interaction, and chose to have each action accompanied by sound (vocalisations). Indeed, vocalising in response to the infant's actions is the most common way in which contingent feedback is realised in mother-infant interaction [30].

Following previous studies on imitation recognition and social contingency discrimination, each experimental condition was followed by a brief still-face (henceforth SF) interval [20,21]. The duration of the SF interval was set to 30 seconds, as pilot testing indicated this to be sufficient for the purposes of this study, i.e. for eliminating carryover effects from an experimental condition to another without causing too much distress to the infants. Accordingly, these were the criteria used when establishing the duration of the still-face intervals. As such, we sought to avoid an unnecessary prolongation of the SF intervals, as both pilot testing and previous studies indicated that this would jeopardize testing. Firstly, long SF intervals risk to cause too much distress in the infants, and require long recovery times once the mother or experimenter becomes active again [39]. Secondly, the overall duration of the experimental phase would also increase, thus becoming too straining for such young infants. As a consequence, based on pilot testing and findings from previous studies, and given the specific purposes of this study, the duration of an SF interval was shorter than the duration of each experimental condition. This differs from studies aimed at assessing the SF effect in relation to e.g. regular mother-infant interaction, in which the duration of the SF interval equals the duration of the regular interaction. Assessing the SF effect, however, goes beyond the purposes of the present study, as stated further above. Nevertheless, analysing the infants' responses during the SF intervals could be potentially revealing, as it has been speculated that these could provide an additional measure on the effects of condition. This speculation is based on findings suggesting that infants' social responsiveness during SF intervals is susceptible to effects of previous engagements, at least in the long term. For example, it has been found that infants whose mothers exhibit higher levels of mirroring exhibit more *social bidding* during SF [3,24]. *Social bidding*—which has been operationalised as non-distress vocalisations [3,25] or gestures such as banging, clapping, etc. directed at the interaction partner [20]—is considered to be an indicator of the infants' expectations for social engagement and their attempt to elicit a response from the interaction partner [3]. A previous study on imitation recognition with older infants (aged 9-, 14-, and 18-months old), however, failed to find any differences with respect to social biddings directed toward an imitating compared to a contingently responsive experimenter [20]. Besides levels of social bidding, levels of attention and smiling were also similar in this study.

**Debriefing.** Upon test completion the mother was debriefed regarding the study, and the infant was handed the toy.

## Measured behaviours and data coding

Following previous studies on imitation recognition in infants [4,6,11,12,13,20], *implicit imitation recognition* was assessed by quantifying infants' social responsiveness during each of the four experimental conditions. This assessment included the two measures of social responsiveness that are customarily employed in studies on infant imitation recognition and effects of social contingency—i.e. visual attention and smiling [4,5,6,11,12,13,20–22,37] -, as well as a novel measure, i.e. approach attempts directed toward the experimenter. For each of these three behavioural measures of social responsiveness, we quantified: (i) overall duration; (ii) number of bouts; (iii) average bout duration.

An attention bout was coded for as long as the gaze of the infant was oriented toward the experimenter's face or relevant body part, i.e. the imitating / contingently responsive hand, arm, etc. A smile was coded whenever the infant's lips were stretched with lip corners oriented upwards compared to a reference neutral expression. Open mouth smiles—with or without visible gums—and laughter were also included in this category. An additional—but optional—criterion for smiling was eye constriction, which helped disambiguating in situations when visibility of the infant's mouth was obstructed by e.g. a pacifier or the infant's hands. Only smiles directed toward the experimenter (face, relevant body part) were scored. Smiles directed toward the parent, reflecting surfaces, objects, while the infant was not attending to the experimenter, were not scored for the purposes of this study. Approach behaviours included infants' attempts to come closer to the experimenter, such as leaning toward the experimenter above or behind the table, climbing on the table, or trying to touch the experimenter.

To establish if 6-month old infants exhibited *explicit imitation recognition*, the data was screened for testing behaviours, including behavioural repetitions, behavioural modulations and testing sequences. In line with previous studies [4,5,20], a repetition was scored only if a given action was repeated at least four times. A behavioural modulation was scored whenever a particular action was repeated with variations, such as speeding up or slowing down the action, modifying its amplitude, switching the effector arm, sudden posture freezing and restarts. A testing sequence was scored whenever the infant repeated and alternated two or more actions in response to being imitated. In addition, we also scored if the infant monitored the experimenter's actions during testing, and if a testing bout was accompanied by smiling. The presence of testing behaviours was quantified with respect to frequency and duration. To assess how readily 6-month old infants recognised that they were being imitated, we also quantified latency to the first testing bout.

In addition, following [20], infants' social responsiveness—attention, smiling, approach—was also measured during each SF interval. Testing behaviours were not scored during the SF intervals, as the experimenter remained immobile, and, thus, there were no actions to be subjected to testing by the infants. Instead, we scored attempts to re-engage the experimenter in interaction, i.e. *social bidding*. Descriptions of social bidding vary across studies from including minimally non-distress vocalisations directed toward the experimenter [3,24], to including additional behaviours such as clapping, banging, trying to reach the experimenter [20]. Given the low number of non-distress vocalisations produced during the SF intervals, we opted for the more inclusive approach, and thus a social bidding bout was coded whenever the infant vocalised toward the experimenter, or produced another behaviour directed toward the experimenter.

## Assessing the efficacy of the CR condition

As reviewed above, CR has been differently operationalised in previous studies on imitation recognition with infants as either spontaneous mother-infant interaction or as standardised

CR by an experimenter. It was therefore important to assess the efficacy of the CR condition to the tested age group, by assessing the comparability of the two approaches. To this end, we compared levels of attention and smiling as measured in the experimental group during the CR condition to those measured during mother-infant interaction in a matched control group. Participants in the control group were 15 age-matched infants (mean age = 199.2 days, SD = 5.583, 7 females, 8 males) and their mothers. Data from an additional number of 5 infants could not be included in the study due to equipment malfunction (N = 1), fussiness (N = 3) or poor illumination (N = 1). The infants were visited in their homes, and each visit had a similar structure to that described for the experimental group, with the exception that the experimental phase was replaced by a phase of mother-infant interaction. The mothers were instructed to play or interact with the infants as they usually do for three minutes (after which they had to follow a script, for the purposes of another study). During the mother-infant interaction, the infants were seated either in a high-chair, a rocking chair or on their mother's knees. To match the duration of the CR condition, only 2 minutes of the spontaneous mother-infant interaction were used for comparison purposes. The coding of relevant responses—attention, smiling— was conducted as described in the previous section. Approach behaviours could not be assessed for the control group since proximity in the mother-infant dyad was to a large extent regulated by the mother. In contrast, in the experimental group, infant-experimenter proximity was entirely controlled by the infant.

## Statistical analysis

To determine if 6-month-old infants showed *implicit imitation recognition*, the overall duration, frequency and the average bout duration of social responses (attention, smiling, approach) was compared across conditions. Following previous studies [e.g. 20], the effect of condition on social responsiveness, was assessed by conducting repeated measures ANOVAs followed-up by paired *t*-tests. Whenever parametric test assumptions were violated, data transformation was carried out, and parametric tests were conducted on transformed data if normality was sufficiently improved. If data transformation failed to sufficiently improve normality, Friedman's ANOVAs were instead carried out, followed-up by the Wilcoxon Signed-Rank test. Given the small sample, the Shapiro-Wilk test was employed to determine if data differed significantly from a normal distribution [40]. For the same reason (i.e. a small sample size), exact significance (two-tailed) is always reported for the non-parametric tests [40]. To determine if 6-month old infants showed *explicit imitation recognition*, the effects of condition on the duration, frequency and latency of testing behaviours was assessed using a similar approach as described above for *implicit* measures. These statistical analyses were conducted using SPSS version 25.

To control for false discovery rates whenever conducting multiple comparisons, all *p*-values were corrected using the Benjamini-Hochberg procedure, which is commonly known for striking a good balance between limiting false positives and preventing power reduction. Effect sizes for the parametric ANOVAs are reported as partial eta squared, and were calculated in SPSS. For Friedman's ANOVAs, effect sizes were computed following [41], whereby for Kendall's *W* df (3), an effect size is small if < 0.1, medium if comprised between 0.1–0.3, and large if ≥ 0.3. Effect sizes for the focused comparisons (Cohen's *dz*) were computed in G*Power, following [42].

Preliminary analyses relying on the same methods as described above were conducted in order to assess the effects of sex and presentation order on the 12 dependent variables. Presentation order was treated as a within-subject variable with four levels (i.e. slot 1, slot 2, slot 3, slot 4) and its effect was assessed by conducting repeated measures ANOVAs or Friedman's

ANOVAs, depending on whether the data met parametric test assumptions. Sex was treated as a between-subjects variable, and its effect on attention, smiling, approach and testing behaviours was assessed by conducting independent t-tests or the Mann-Whitney U test. Additional analyses (e.g. mixed-model ANOVAs, or series of two-sample *t*-tests carried out separately for each experimental slot or experimental condition) were also conducted in order to capture potential interaction effects (see S1 Data for details). Mixed-model ANOVAs including all factors (sex, order and condition) could not be carried out since our counterbalanced design was affected by participant exclusion, thus leading to variable—and sometimes too small—order-by-condition sample sizes.

Similar analyses to those described in the previous paragraphs were separately conducted for the SF intervals, in order to assess the effects of order, sex and preceding condition on the dependent variables during the SF intervals. Dependent variables for these analyses were the overall duration, frequency and average bout duration of attention, smiling, approach and social bidding directed toward the experimenter.

### Ethical considerations

The procedure carried out is in accordance with the 1964 Declaration of Helsinki (and its later amendments), and has been approved by the Regional Ethical Review Board Lund (permit 2013/317). The participants were recruited from birth records in Stockholm county (Sweden), provided by the Swedish Population Register (permit 131 520184-14/9121). Informed consent has been obtained from the legal guardians of the infants. For the filmed material that accompanies the study for illustrative purposes, informed consent was given by a parent for the publication of identifying images in an online open-access publication.

## Results

Given the large number of dependent variables, and thus the large number of statistical tests conducted in order to assess the effects of condition (and additional independent variables such as order, sex) on these, only significant results will be reported in the *Results* section of the manuscript (see also Figs 1, 2, 3 and 4). Non-significant results concerning the effect of condition, order and sex are detailed in sections **1–3** of the S1 Data.

### Inter-coder agreement

To establish coding reliability, 30% of the data was scored by a second coder, who was naive about the purposes of the study at the time of coding. Intercoder agreement was found to be excellent for all measured behaviours: *attention*: $r = 0.996$, N = 311, $p < 0.001$; *smiling*: $r = 0.975$, N = 188, $p < 0.001$; *approach*: $r = 0.998$, N = 43, $p < 0.001$; *testing*: $r = 0.854$, N = 72, $p < 0.001$ (all *p*-s two-tailed).

### Preliminary tests for the effect of sex and order during the experimental conditions

To test the effects of order on the 12 dependent variables, six one-way repeated measures ANOVAs and six Friedman's ANOVAs were conducted (depending on whether data met parametric test assumptions). All these 12 tests yielded non-significant results thus indicating that order did not affect any of the 12 dependent variables. In other words, infants' attention, smiling, approach and testing behaviours did not vary between the first, second, third and fourth experimental slot, which could be expected as a result of e.g. boredom, tiredness or carryover effects. Nevertheless, one of the supplementary mixed-model ANOVAs revealed a

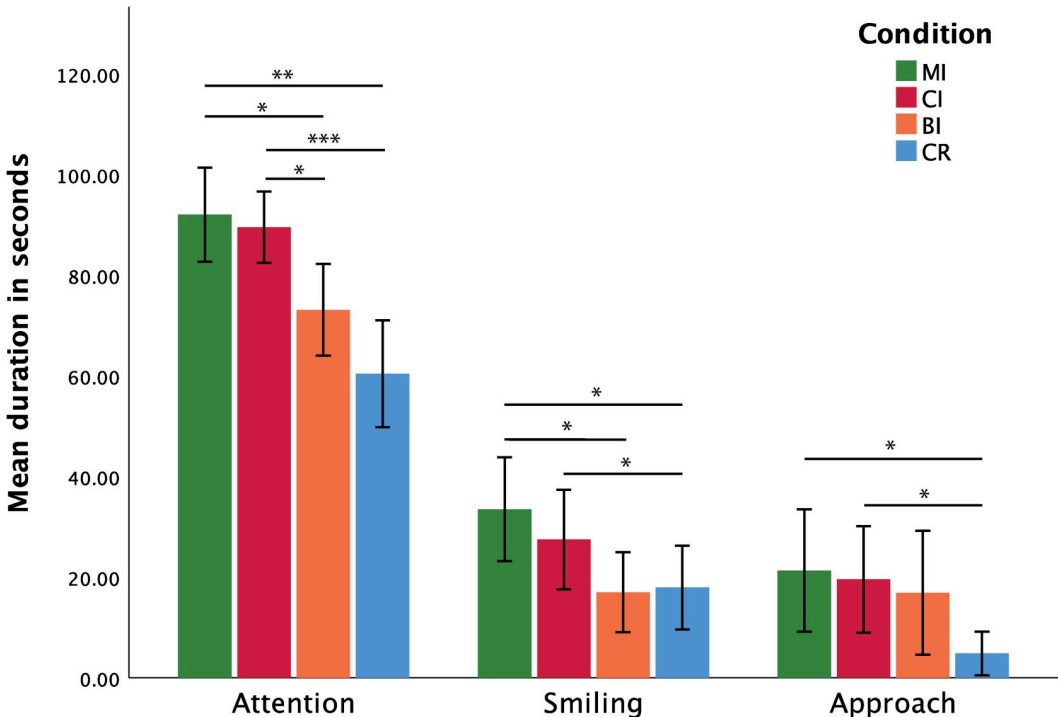

**Fig 1. Overall duration, implicit measures.** Mean duration (and standard error) of attention to experimenter (left), smiling (centre), and approach behaviours (right) during the four experimental conditions (* indicates p < 0.05; ** indicates p < 0.01; *** indicates p < 0.001).

significant interaction effect of order and sex on the frequency of attention bouts: F (3,42) = 3.097, p = 0.037, $\eta_p^2$ = 0.181. Follow-up t-tests were conducted to break down this interaction, but all these tests yielded non-significant results (see section **1** of the S1 Data for details). Means (and standard error of the mean) based on non-transformed data are summarised in Table 1. All the other supplementary mixed-model ANOVAs were non-significant (see section **1** of the S1 Data for additional details).

Likewise, depending on whether data met parametric test assumptions, four independent *t*-tests and eight Mann-Whitney tests were conducted to assess the effects of sex on the dependent variables. All these tests yielded non-significant results, with the exception of the t-test conducted to assess the effect of sex on the average duration of smiling bouts, which was significant: t(62) = 2.737, p = 0.008, d = 0.744. Follow-up independent t-tests were carried out to assess the effects of sex on the average duration of smiling bouts separately, for each order slot and each condition. The results yielded by all these additional tests were non-significant (see section **2** of the S1 Data for details). Similar supplementary tests (i.e. separately for each experimental slot and each experimental condition) were conducted for each variable, but all these tests yielded non-significant results. Means (and standard error of the mean) based on non-transformed data are summarised in Table 2.

## Implicit imitation recognition

**Effects of condition on attention.**    For the *total duration of attention*, the Shapiro-Wilk tests indicated that the normality assumption was violated for the MI and CR conditions: $W_{MI}$ = 0.854, p = 0.016; $W_{CR}$ = 0.894, p = 0.026. Since normality was not improved by data transformation, a Friedman's ANOVA was conducted to assess the effects of condition on the *total*

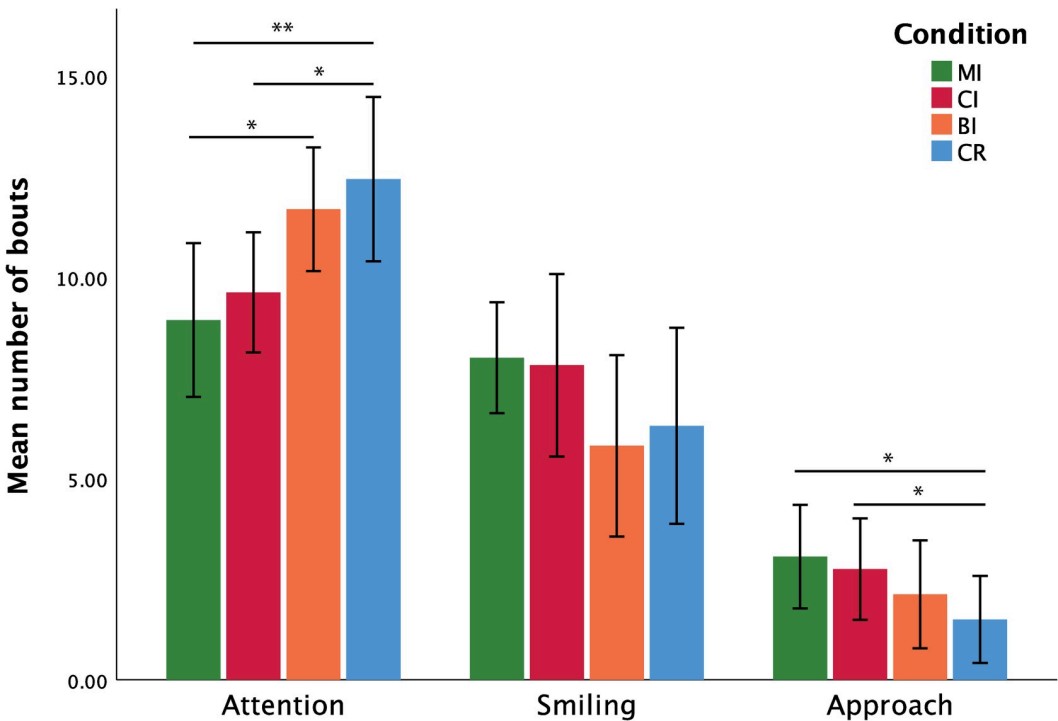

**Fig 2. Frequency, implicit measures.** Mean number of bouts (and standard error) for attention to experimenter (left), smiling (centre), and approach behaviours (right) during the four experimental conditions (* indicates $p < 0.05$; ** indicates $p < 0.01$; *** indicates $p < 0.001$).

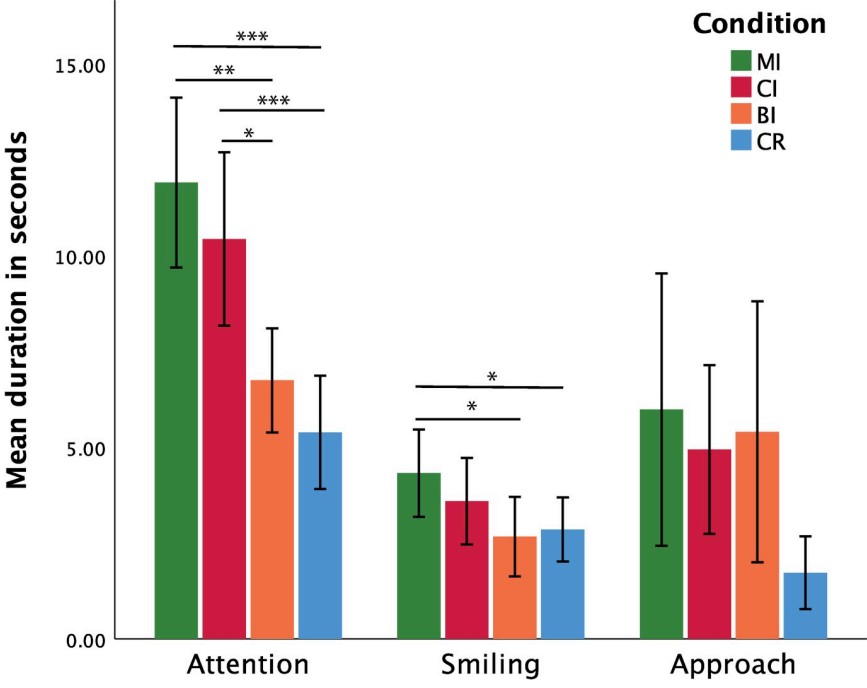

**Fig 3. Average bout duration, implicit measures.** Mean bout duration (and standard error) for attention to experimenter (left), smiling (centre), and approach behaviours (right) during the four experimental conditions (* indicates $p < 0.05$; ** indicates $p < 0.01$; *** indicates $p < 0.001$).

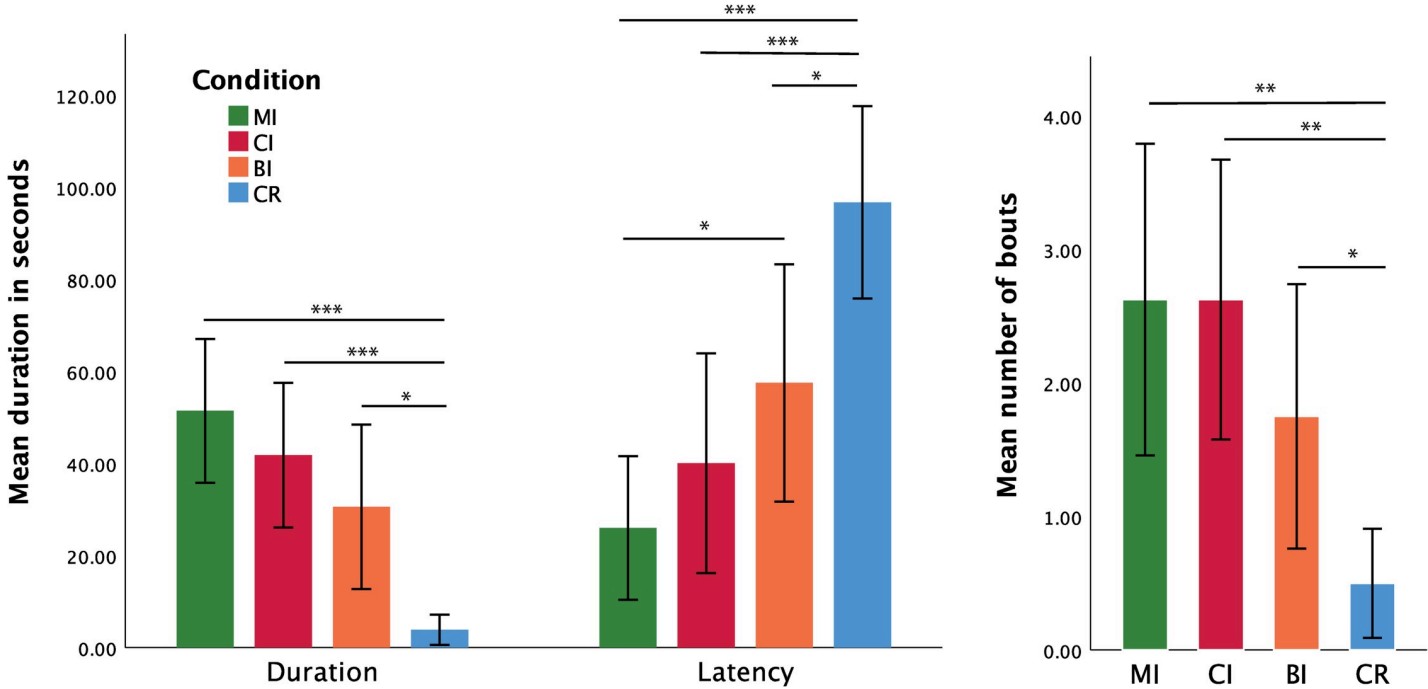

**Fig 4. Testing behaviours.** Mean duration (and standard error) of testing behaviours (to the left), latency to the first testing bout (centre), and frequency of testing behaviours (to the right) in the four experimental conditions (* indicates $p < 0.05$; ** indicates $p < 0.01$; *** indicates $p < 0.001$).

*duration of attention*. The results of this test were significant: $\chi^2(3) = 25.575$, $p < 0.001$, $W = 0.533$. Follow-up analyses using the Wilcoxon Signed-Rank test indicated that attention lasted longer in the MI and CI conditions compared to the BI and CR conditions (MI vs. BI: $Z = -2.689$. $p = 0.010$; $dz = 0.786$; MI vs. CR: $Z = -2896$, $p = 0.006$; $dz = 1.021$; CI vs. BI: $Z = -2.275$, $p = 0.032$, $dz = 0,683$; CI vs. CR: $Z = -3.516$, $p < 0.001$, $dz = 1.407$).

**Table 1. Mean (± s.e.m.) duration and frequency for measured responses in the four experimental slots, i.e. as a function of presentation order.**

| Experimental slot | Slot 1 | Slot 2 | Slot 3 | Slot 4 |
|---|---|---|---|---|
| **Dependent variable** | | | | |
| **Attention** | | | | |
| Duration (in sec) | 86.143 (5.309) | 73.323 (4.905) | 71.865 (5.961) | 83.830 (5.405) |
| Frequency (no. bouts) | 10.125 (0.752) | 11.875 (1.016) | 10.688 (0.865) | 10.000 (1.076) |
| Avg bout duration (in sec) | 9.655 (1.207) | 7.293 (1.001) | 7.706 (1.076) | 9.812 (1.194) |
| **Smiling** | | | | |
| Duration (in sec) | 27.437 (4.220) | 16.984 (3.831) | 21.621 (3.498) | 29.928 (6.746) |
| Frequency (no. bouts) | 8.625 (1.245) | 5.438 (0.966) | 6.688 (0.956) | 7.188 (1.001) |
| Avg bout duration (in sec) | 3.035 (0.393) | 3.058 (0.517) | 3.496 (0.455) | 3.855 (0.737) |
| **Approach** | | | | |
| Duration (in sec) | 17.470 (5.032) | 13.300 (4.987) | 12.765 (4.580) | 19.129 (6.741) |
| Frequency (no. bouts) | 2.688 (0.617) | 2.438 (0.639) | 1.563 (0.508) | 2.750 (0.733) |
| Avg bout duration (in sec) | 4.787 (1.038) | 3.806 (1.002) | 5.168 (2.051) | 4.293 (1.362) |
| **Testing behaviours** | | | | |
| Duration (in sec) | 38.864 (8.401) | 24.282 (7.611) | 31.037 (8.682) | 33.491 (8.857) |
| Frequency (no. bouts) | 2.188 (0.530) | 1.625 (0.437) | 1.625 (0.473) | 2.063 (0.623) |
| Latency to first bout (in sec) | 52.688 (13.599) | 56.438 (12.152) | 56.438 (12.971) | 54.750 (12.835) |

**Table 2. Mean (± s.e.m.) duration and frequency for measured responses as a function of sex.**

| Sex | Males | Females |
|---|---|---|
| **Dependent variable** | | |
| **Attention** | | |
| Duration (in sec) | 78.087 (3.307) | 80.338 (5.088) |
| Frequency (no. bouts) | 10.432 (0.451) | 11.200 (1.128) |
| Avg bout duration (in sec) | 8.475 (0.649) | 8.927 (1.137) |
| **Smiling** | | |
| Duration (in sec) | 26.947 (3.055) | 17.492 (3.403) |
| Frequency (no. bouts) | 6.909 (0.591) | 7.150 (1.122) |
| Avg bout duration (in sec) | 3.814 (0.346) | 2.365 (0.296) |
| **Approach behaviours** | | |
| Duration (in sec) | 15.483 (3.260) | 16.070 (4.686) |
| Frequency (no. bouts) | 2.523 (0.394) | 2.000 (0.508) |
| Avg bout duration (in sec) | 4.311 (0.835) | 4.957 (1.304) |
| **Testing behaviours** | | |
| Duration (in sec) | 33.099 (4.997) | 29.322 (7.620) |
| Frequency (no. bouts) | 1.773 (0.245) | 2.100 (0.628) |
| Latency to first bout (in sec) | 51.455 (7.287) | 63.050 (12.310) |

For the *frequency of attention bouts*, the Shapiro-Wilk tests indicated that the normality assumption was violated for the MI and CR conditions: $W_{MI} = 0.854$, $p = 0.016$; $W_{CR} = 0.894$, $p = 0.026$. Logarithmic transformation improved data normality: $W_{MI} = 0.954$, $p = 0.548$; $W_{CR} = 0.929$, $p = 0.235$. A repeated-measures ANOVA was thus conducted on the log transformed data in order to assess the effects of condition on the *frequency of attention bouts*. Mauchly's Test of Sphericity indicated that the sphericity assumption was met ($\chi^2(5) = 1.540$; $p = 0.909$). The repeated-measures ANOVA yielded significant results: $F(3,45) = 6.059$, $p = 0.001$, $\eta_p^2 = 0.288$. Follow-up comparisons based on paired t-tests (also conducted on log transformed data) showed that attention disengagement was significantly less frequent in the MI and CI conditions compared to the CR condition (MI vs. CR: $t(15) = -3.937$, $p = 0.006$, $dz = 0.980$; CI vs. CR: $t(15) = -2.777$, $p = 0.028$, $dz = 0.691$) and in the MI condition compared to the BI condition ($t(15) = -2.924$, $p = 0.028$, $dz = 0.730$).

For the *average duration of attention* bouts, the Shapiro-Wilks tests indicated that the normality assumption was violated for the CI and BI conditions: $W_{CI} = 0.858$, $p = 0.018$; $W_{BI} = 0.785$, $p = 0.002$. As data transformation did not improve normality, a Friedman's ANOVA was conducted to assess the effects of condition on the *average duration of attention bouts*. The results of this test were significant: $\chi^2(3) = 25.875$, $p < 0.001$, $W = 0.533$. Follow-up analyses using the Wilcoxon Signed-Rank test revealed that *average bout duration* was longer in the MI and CI conditions compared to the BI and CR conditions (MI vs. BI: $Z = -2.741$, $p = 0.008$, $dz = 0,905$; MI vs. CR: $Z = -3.309$, $p < 0.001$, $dz = 1.327$; CI vs. BI: $Z = -2.430$, $p = 0.019$; $dz = 0.644$; CI vs. CR: $Z = -3.258$, $p < 0.001$, $dz = 1.039$). As a reminder, note that, for all multiple comparison reported throughout this section, p-values have been corrected for false discovery rates (Type I error) using the Benjamini-Hochberg procedure. Means (and standard error of the mean) based on non-transformed data are summarised in Table 3. Non-significant results concerning attention measures can be consulted in the section **3.1.** of the S1 Data.

**Effects of condition on smiling.** For the *total duration of smiling*, the Shapiro-Wilk tests indicated that the normality assumption was violated for the BI condition: $W_{BI} = 0.877$, $p = 0.035$. Logarithmic transformation, however, improved data normality ($W_{BI} = 0.912$,

**Table 3. Mean (± s.e.m.) duration and frequency for measured responses in the four experimental conditions.**

| Condition<br>Dependent variable | Mirror imitation | Contralateral imitation | Bodily imitation | Contingent responding |
|---|---|---|---|---|
| **Attention** | | | | |
| Duration (in sec) | 92.046 (4.683) | 89.555 (3.546) | 73.124 (4.555) | 60.437 (5.308) |
| Frequency (no bouts) | 8.938 (0.955) | 9.625 (0.747) | 11.688 (0.768) | 12.438 (1.020) |
| Avg bout duration (in sec) | 11.903 (1.109) | 10.433 (1.130) | 6.743 (0.677) | 5.387 (0.739) |
| **Smiling** | | | | |
| Duration (in sec) | 33.517 (5.166) | 27.484 (4.954) | 17.026 (3.987) | 17.945 (4.166) |
| Frequency (no bouts) | 8.000 (0.689) | 7.813 (1.134) | 5.813 (1.126) | 6.313 (1.217) |
| Avg bout duration (in sec) | 4.324 (0.268) | 3.593 (0.565) | 2.671 (0.518) | 2.857 (0.418) |
| **Approach** | | | | |
| Duration (in sec) | 21.333 (6.091) | 19.571 (5.282) | 16.924 (6.147) | 4.836 (2.167) |
| Frequency (no bouts) | 3.063 (0.642) | 2.750 (0.629) | 2.125 (0.670) | 1.500 (0.540) |
| Avg bout duration (in sec) | 5.982 (1.775) | 4.942 (1.101) | 5.402 (1.702) | 1.726 (0.473) |
| **Testing behaviours** | | | | |
| Duration (in sec) | 51.439 (7.815) | 41.787 (7.849) | 30.581 (8.931) | 3.868 (1.649) |
| Frequency (no bouts) | 2.625 (0.584) | 2.625 (0.523) | 1.75 (0.496) | 0.5 (0.204) |
| Latency to first bout (in sec) | 26.000 (7.801) | 40.063 (11.940) | 57.500 (12.893) | 96.750 (10.441) |

p = 0.125). A repeated-measures ANOVA was thus conducted on the log transformed data to assess the effects of condition on the *total duration of smiling*. Mauchly's Test of Sphericity indicated that the assumption of sphericity was not met: $\chi^2(5) = 13.448$; p = 0.020. Degrees of freedom were therefore corrected using the Greenhouse-Geisser estimate of sphericity ($\varepsilon = 0.700$). The results of the ANOVA were significant: F (2.11; 31.5) = 5.777, p = 0.007, $\eta_p^2 = 0.278$. Follow-up comparisons using the paired t-test (based on log transformed data) revealed that smiling lasted longer in the MI and CI conditions compared to the CR condition, and in the MI condition compared to the BI condition (MI vs. BI: t(15) = 3.252, p = 0.021, dz = 0.812; MI vs CR: t(15) = 2.913, p = 0.022, dz = 0.727; CI vs. CR: t(15) = 3.100, p = 0.021, dz = 0.775).

For the *average duration of smiling bouts*, the Shapiro-Wilk tests indicated that the normality assumption was violated for the CI and CR conditions: $W_{CI} = 0.687$, p < 0.001; $W_{CR} = 0.774$, p = 0.001. Logarithmic transformation improved data normality: $W_{CI} = 0.893$, p = 0.062; $W_{CR} = 0.920$, p = 0.172. A repeated-measures ANOVA was thus conducted to assess the effects of condition on *the average duration of smiling bouts*. Mauchly's Test of Sphericity further showed that the sphericity assumption was met: $\chi^2(5) = 7.638$; p = 0.178. The results of the ANOVA were significant: F(3,45) = 4.645, p = 0.007, $\eta_p^2 = 0.236$. Follow-up paired t-tests (conducted on log transformed data) showed that smiling bouts lasted significantly longer in the MI condition compared to the BI and CR conditions (MI vs. BI: t(15) = 2.926, p = 0.030, dz = 0.730; MI vs. CR: t(15) = 3.119, p = 0.030, dz = 0.779).

As previously, for all multiple comparisons reported throughout this section, p-values have been corrected for false discovery rates (Type I error) using the Benjamini-Hochberg procedure. The results of the test conducted to assess the effects of condition on the *frequency of smiling bouts* are reported only in the S1 Data (section **3.2.2.**), as these were non-significant. Additional non-significant results concerning smiling measures can be consulted in the section **3.2.** of the S1 Data. Means (and standard error of the mean) based on non-transformed data are summarised in Table 3.

**Effects of condition on approach.** For the *total duration of approach* behaviours, the Shapiro-Wilk tests indicated that the normality assumption was violated for all four conditions:

$W_{MI}$ = 0.828, p = 0.007; $W_{CI}$ = 0.851, p = 0.014; $W_{BI}$ = 0.706, p < 0.001; $W_{CR}$ = 0.600, p < 0.001. Since normality was not improved by data transformation, a Friedman's ANOVA was conducted to assess the effects of condition on the *total duration of approach* behaviours. This test revealed a significant effect of condition: $\chi^2$(3) = 8.511, p = 0.034, W = 0.177. Follow-up tests using the Wilcoxon Signed-Rank test showed that approach duration was significantly longer during the MI and CI conditions compared to the CR condition (MI vs. CR: Z = -2.551, p = 0.033, dz = 0.678; CI vs. CR: Z = -2.691, p = 0.033, dz = 0.778).

For the *frequency of approach bouts*, the Shapiro-Wilk tests indicated that the normality assumption was violated for three of the four conditions: $W_{MI}$ = 0.919, p = 0.162; $W_{CI}$ = 0.878, p = 0.036; $W_{BI}$ = 0.776, p = 0.001; $W_{CR}$ = 0.730, p < 0.001. Since normality was not improved by data transformation, a Friedman's ANOVA was conducted to assess the effects of condition on *approach frequency*. The results of this test were significant: ($\chi^2$(3) = 9.585, p = 0.019, W = 0.200). Follow-up tests using the Wilcoxon Signed-Rank test indicated that approach frequency was significantly higher during the MI and CI conditions compared to the CR condition (MI vs CR: Z = -2.540, p = 0.048, dz = 0.735; CI vs. CR: Z = -2.431, p = 0.048, dz = 0.723).

As previously, all multiple comparison p-values presented throughout this section have been corrected for false discovery rates (Type I error) using the Benjamini-Hochberg procedure. The results of the test conducted to assess the effects of condition on the *average duration of approach bouts* are reported only in the S1 Data (section **3.3.3.**), as these were non-significant. Additional non-significant results concerning approach measures can be consulted in the section **3.3.** of the S1 Data. Means (and standard error of the mean) based on non-transformed data are summarised in Table 3.

## Explicit imitation recognition: the effects of condition on testing behaviours

For the *total duration of testing behaviours*, Shapiro-Wilk tests indicated that the normality assumption was violated for two of the four conditions: $W_{MI}$ = 0.960, p = 0.667; $W_{CI}$ = 0.942, p = 0.375; $W_{BI}$ = 0.806, p = 0.003; $W_{CR}$ = 0.666, p < 0.001. As normality was not improved by data transformation, a Friedman's ANOVA was conducted to assess the effects of condition on the *total duration of testing behaviours*. The result of this test was significant: $\chi^2$(3) = 24.890, p < 0.001. Follow-up paired comparisons using the Wilcoxon Signed-Rank test showed that testing behaviours lasted significantly longer during each of the imitation conditions compared to the CR condition (MI vs. CR: Z = -3.408, p < 0.001, dz = 1.577; CI vs. CR: Z = -3.180, p < 0.001, dz = 1.266; BI vs. CR: Z = -2.490, p = 0.02, dz = 0.762).

For the *frequency of testing bouts*, the Shapiro-Wilk tests indicated that the normality assumption was violated for all four conditions: $W_{MI}$ = 0.765, p = 0.001; $W_{CI}$ = 0.870, p = 0.027; $W_{BI}$ = 0.811, p = 0.004; $W_{CR}$ = 0.644, p < 0.001. As normality was not improved by data transformation, a Friedman's ANOVA was conducted to assess the effects of condition on the *frequency of testing bouts*. The result of this test was significant ($\chi^2$(3) = 18.044, p < 0.001), and follow-up paired comparisons using the Wilcoxon Signed-Rank indicated that testing frequency was significantly higher during each of the imitation conditions compared to the CR condition: MI vs. CR: Z = -3.107, p = 0.003, dz = 0.889; CI vs. CR: Z = -3.075, p = 0.001, dz = 1.083; BI vs. CR: Z = -2.359, p = 0.042, dz = 0.600.

For *latency to the first testing bout*, the Shapiro-Wilk tests indicated that the normality assumption was violated for all four conditions: $W_{MI}$ = 0.727, p < 0.001; $W_{CI}$ = 0.718, p < 0.001; $W_{BI}$ = 0.767, p = 0.001; $W_{CR}$ = 0.611, p < 0.001. As normality was not improved by data transformation, a Friedman's ANOVA was conducted to assess the effects of condition on *latency to the first testing bout*. The result of this test was significant ($\chi^2$(3) = 18.600,

p < 0.001), and follow-up paired comparisons using the Wilcoxon Signed-Rank test showed that infants were significantly faster at detecting that they were being imitated in all the imitation conditions compared to the CR condition (MI vs. CR: Z = -3.239, p < 0.001, dz = 1.449; CI vs. CR: Z = -3.180, p < 0.001, dz = 1.135; BI vs CR: Z = -2.224, p = 0.036, dz = 0.730), but also in the MI condition compared to the BI condition: Z = - 2.301, p = 0.036, dz = 0.680.

Finally, to assess if there was an effect of condition on the *proportion of children* who engaged in testing behaviours, a Cochran's Q test was conducted, which revealed significant differences between the four conditions: χ2(3) = 15.333, p = 0.002. Follow-up comparisons using the McNemar test revealed that the proportions of infants who engaged in testing behaviours was significantly greater during the MI and CI conditions compared to the CR condition (p = 0.004, and p = 0.016 respectively).

As previously, for all the multiple comparisons reported in this section, p-values have been corrected using the Benjamini-Hochberg procedure in order to control for Type I error. There were no other significant effects for testing measures (see also Fig 4). Additional non-significant results can be consulted in the S1 Data, section **3.4.**

The most frequent form of testing behaviours exhibited by the infants in response to being imitated was *behavioural repetition*, which was present in 95% (N = 114) of the 120 testing bouts identified during the imitation conditions. The majority of testing bouts (68%, N = 82 bouts), however, were mixed bouts, which also contained more complex forms of testing, such as *behavioural modulations* (present in 40% of the bouts, N = 48) and *testing sequences* (present in 55% of the bouts, N = 66). Smiling accompanied testing bouts in 71% of the cases (N = 85 bouts). The 7 bouts of testing behaviours shown during the CR condition were exclusively behavioural repetitions.

## The efficacy of the contingent responding condition

No significant differences were found between the experimental group during the CR condition and the control group with respect to *duration of attention* (t(29) = 0.314, p = 0.756), *frequency of attention* bouts (t(29) = 0.775, p = 0.444) or *average duration of attention* bouts (U = 119, p = 0.984, two-tailed). Moreover, there was no significant difference between the two groups with respect to *smiling duration* (t(29) = 0.662, p = 0.513), *frequency of smiling* bouts (t(29) = 0.444, p = 0.661) or *average duration of smiling* bouts (t (29) = 0.929, p = 0.361). Please note that the t-tests reported above were all conducted on log-transformed data, as the normality assumption was initially violated. Means (and standard error of the mean) based on non-transformed data are presented in Table 4.

**Table 4. Mean (± s.e.m.) for attention and smiling (total duration, frequency, average bout duration) during the contingent responding condition (CR) compared to a mother-infant interaction (control group).**

| Group | Experimental CR condition | Control Mother-infant interaction |
|---|---|---|
| **Dependent variable** | | |
| **Attention** | | |
| Duration (in sec) | 60.437 (5.308) | 57.955 (5.879) |
| Frequency (no. bouts) | 12.438 (1.021) | 12.065 (1.355) |
| Avg bout duration (in sec) | 5.387 (0.739) | 6.839 (1.963) |
| **Smiling** | | |
| Duration (in sec) | 17.945 (4.166) | 17.132 (4.447) |
| Frequency (no. bouts) | 6.313 (1.217) | 6.000 (1.254) |
| Avg bout duration (in sec) | 2.857 (0.418) | 2.533 (0.504) |

## The effects of order, sex and condition on the dependent variables during the SF intervals

**Effects of order on attention, smiling, approach, and social bidding.**   Data analysis revealed order effects on attention (overall duration and average bout duration), smiling (overall duration, frequency, and average bout duration), and social bidding (frequency). With respect to *overall duration of attention bouts*, Shapiro-Wilk tests showed that the normality assumption was met for all datasets: $W_{SF1}$ = 0.947, p = 0.448; $W_{SF2}$ = 0.951, p = 0.500; $W_{SF3}$ = 0.923, p = 0.191; $W_{SF4}$ = 0.931, p = 0.257. A repeated-measures ANOVA was thus conducted to assess the effects of order on the duration of attention during the SF intervals. Mauchly's test indicated that the sphericity assumption was met: $\chi^2(5)$ = 4.265; p = 0.513. The results of the ANOVA were significant: F(3,45) = 10.232, p < 0.001, $\eta_p^2$ = 0.406. Follow-up analyses conducted using the paired t-test revealed that the overall duration of attention was longer during the first SF interval compared to each subsequent SF interval. As such, SF-1 vs. SF-2: $t(15)$ = 3.653, p = 0.006 (uncorrected p = 0.002), dz = 0.913; SF-1 vs. SF-3: $t(15)$ = 3.569, p = 0.006 (uncorrected p = 0.003), dz = 0.892; SF-1 vs. SF-4: $t(15)$ = 4.818, p < 0.001 (uncorrected p < 0.001), dz = 1.204. No other comparisons yielded significant results with respect to the effects of order on the overall duration of attention bouts during the SF intervals, and there was no significant interaction effect between order and sex (see S1 Data, section **4.1.1.** for additional details).

For the *average duration of attention bouts* during the SF intervals, Shapiro-Wilk tests showed that the normality assumption was met for all datasets: $W_{SF1}$ = 0.901, p = 0.084; $W_{SF2}$ = 0.903, p = 0.090; $W_{SF3}$ = 0.980, p = 0.960; $W_{SF4}$ = 0.967, p = 0.665. A repeated-measures ANOVA was thus conducted to assess the effects of order on the average duration of attention bouts during the SF intervals. Mauchly's test showed that the sphericity assumption was met: $\chi^2(5)$ = 3.467, p = 0.871. The ANOVA yielded significant results: F (3,45) = 6.602, p = 0.001, $\eta_p^2$ = 0.306. Follow-up comparisons based on the paired t-test revealed that the average duration of attention was higher in the first SF interval compared to each of the subsequent SF intervals. As such, SF-1 vs. SF-2: $t(15)$ = 2.960, p = 0.02 (uncorrected p = 0.010), dz = 0.740; SF-1 vs. SF-3: $t(15)$ = 3.526, p = 0.018 (uncorrected p = 0.003), dz = 0.882; SF-1 vs. SF-4: $t(15)$ = 3.593, p = 0.009 (uncorrected p = 0.003), dz = 0.898. There were no additional significant differences, and there was no significant interaction effect of sex and order (for non-significant results, see the S1 Data file, section **4.1.3.**). Order did not affect the frequency of attention bouts during the SF intervals, as revealed by the non-significant results reported in the S1 Data file, section **4.1.2.**

For the *overall duration of smiling bouts*, Shapiro-Wilk tests revealed that the normality assumption was severely violated for all datasets $W_{SF1}$ = 0.848, p = 0.013; $W_{SF2}$ = 0.723, p < 0.001; $W_{SF3}$ = 0.670, p < 0.001; $W_{SF4}$ = 0.352, p < 0.001. Data transformation did not sufficiently improve normality, and thus a Friedman's ANOVA was conducted to assess the effects of order on the overall duration of smiling during the SF intervals. The results of this test were significant: $\chi^2(3)$ = 13.703, p = 0.002. Follow-up comparisons using the Wilcoxon Signed-Rank test revealed that the overall duration of smiling was significantly longer during the first SF interval compared to each of the subsequent SF intervals. As such SF-1 vs. SF-2: Z = -2.490, p = 0.03 (uncorrected p = 0.010), dz = 0.607; SF-1 vs. SF-3: Z = -2.341, p = 0.034 (uncorrected p = 0.017), dz = 0.631; SF-1 vs. SF-4: Z = -2.845, p = 0.012 (uncorrected p = 0.002), dz = 0.774. No other comparisons yielded significant results (see the S1 Data file, section **4.2.1.**).

For the *frequency of smiling bouts* during the SF intervals, Shapiro-Wilk tests showed that the normality assumption was violated for all data sets. As such, $W_{SF1}$ = 0.857, p = 0.018; $W_{SF2}$

= 0.697, p < 0.001; $W_{SF3}$ = 0.751, p = 0.001; $W_{SF4}$ = 0.398, p < 0.001. Data transformation could not sufficiently improve normality, and thus a Friedman's ANOVA was conducted to assess the effects of order on the frequency of smiling bouts. This test yielded significant results: $\chi^2$(3) = 13.212, p = 0.002. Follow-up comparisons using the Wilcoxon Signed-Rank test revealed that the frequency of smiling was higher during the first SF interval compared to the last SF interval: Z = -2.682, p = 0.024 (uncorrected p = 0.004), dz = 0.877. No other comparisons yielded significant results (for these additional results see the S1 Data file, section **4.2.2.**).

For the *average duration of smiling bouts* during the SF intervals, Shapiro-Wilk tests revealed that the normality assumption was violated for three datasets: $W_{SF1}$ = 0.890, p = 0.056; $W_{SF2}$ = 0.750, p = 0.001; $W_{SF3}$ = 0.767, p = 0.001; $W_{SF4}$ = 0.352, p < 0.001. As data transformation did not improve normality, the effects of order on the average duration of smiling bouts during the SF intervals was assessed by conducting a Friedman's ANOVA. The results of this test were significant: $\chi^2$(3) = 14.838, p = 0.001. Follow-up comparisons based on the Wilcoxon Signed-Rank test revealed that the average duration of smiling bouts was significantly longer during the first SF interval compared to the second and fourth SF interval. As such, SF-1 vs. SF-2: Z = -2.934, p = 0.006 (uncorrected p = 0.001), dz = 0.985; SF-1 vs. SF-4: Z = -2.845, p = 0.006 (uncorrected p = 0.002), dz = 0.993. No other significant differences were revealed by these comparisons (see the S1 Data file, section **4.2.3.** for additional details), although the comparison between SF-1 and SF-3 approached significance: Z = -2.201, p = 0.054 (uncorrected p = 0.027).

Data analyses revealed that order did not affect the duration or frequency of approach bouts, nor the overall or average duration of social bidding bouts (see the S1 Data file, section **4.3.1.-4.3.3.**, as well as **4.4.1.** and **4.4.3.** for additional details). There was, however, an effect on the *frequency of social bidding bouts*. As Shapiro-Wilks tests showed that the normality assumption was violated for two data sets ($W_{SF1}$ = 0.837, p = 0.009; $W_{SF2}$ = 0.927, p = 0.218; $W_{SF3}$ = 0.887, p = 0.050; $W_{SF4}$ = 0.808, p = 0.003), and data transformation did not improve normality, a Friedman's ANOVA was conducted to assess these effects. This test yielded significant results: $\chi^2$(3) = 11.167, p = 0.008. Follow-up comparisons using the Wilcoxon Sign-Rank test revealed that infants initiated social bidding more often in the first compared to the fourth SF phase: Z = -3.096, p = 0.006 (uncorrected p = 0.001). All other comparisons were non-significant (see the S1 Data file, section **4.4.2** for additional results).

**Effects of sex on attention, smiling, approach, and social bidding.** For the SF intervals, statistical analyses revealed that sex had an effect on the overall duration of attention during the SF intervals, but that it did not affect the other dependent variable (see the S1 Data file, sections **5.1.2.-5.4.3.** for non-significant results). Since Shapiro-Wilk tests revealed that the normality assumption was violated for both datasets ($W_{Males}$ = 0.932, p = 0.013; $W_{Females}$ = 0.886, p = 0.023), data transformation was conducted, with square root transformation leading to sufficiently improved normality: $W_{Males}$ = 0.978, p = 0.553; $W_{Females}$ = 0.969, p = 0.728. An independent t-test conducted on square root transformed data revealed a significant effect of sex on the overall duration of attention during the SF intervals: $t$(14) = 2.204, p = 0.031, d = 0.594. The overall duration of attention of female infants ($M_{Females}$ = 8.813, SE = 1.391) was thus significantly longer than that of male infants ($M_{Males}$ = 5.793, SE = 0.676).

The effect of sex on the total duration of attention was further assessed separately, as a function of order and condition. These tests revealed that, during the first SF interval, there was a significant difference between male and female infants ($t$(14) = 2.515, p = 0.025, d = 1.357), as female infants attended to the experimenter for significantly longer ($M_{Females}$ = 16.188, SE = 3.175) compared to male infants ($M_{Males}$ = 8.942, SE = 1.341). In addition, there was a significant difference between male and female infants in the SF subsequent to the CI condition

($t$(14) = 2.904, p = 0.012, d = 1.156), with female infants attending to the experimenter for significantly longer ($M_{Females}$ = 14.964, SE = 3.487) compared to male infants ($M_{Males}$ = 5.866, SE = 1.444). All other tests yielded non-significant results, as reported in the S1 Data file, section **5.1.1**.

**Effects of condition on attention, smiling, approach, and social bidding.** Statistical analyses revealed no effects of condition on attention, smiling, approach and social bidding during the SF intervals (for details, see the S1 Data file, sections **6.1.1. - 6.4.3.**). There was, however, an interaction effect of condition and sex, which affected the overall duration of attention in the SF intervals. Since the normality assumption was met for all datasets ($W_{SF-MI}$ = 0.938, p = 0.326; $W_{SF-CI}$ = 0.892, p = 0.060; $W_{SF-BI}$ = 0.979, p = 0.956; $W_{SF-CR}$ = 0.922, p = 0.181), this interaction effect was obtained while conducting a mixed- model ANOVA with condition as the within-subjects factor and sex as the between-subjects factor. Mauchly's Test of Sphericity showed that the sphericity assumption was met for this test ($\chi^2$(5) = 4.154; p = 0.528). The results of this test showed a significant main effect of condition (F(3,42) = 3.193, p = 0.033, $\eta^2_p$ = 0.186) and a significant interaction effect between condition and sex: F (3,42) =, 2.938, p = 0.044, $\eta^2_p$ = 0.173). To follow-up on the effect of condition, paired comparisons using the paired t-test were conducted, but all results were non-significant. To follow-up on the significant interaction effect between sex and condition, independent t-tests were conducted separately for each condition dataset, with sex as the between-subjects variable. One of these tests revealed a significant effect of sex for the duration of attention in the SF phase following the CI condition, which is the same significant interaction captured in the previous paragraph: t(14) = 2.904, p = 0.012, d = 1.566 ($M_{Males}$ = 5.866, SE = 1.444; $M_{Females}$ = 14.964, SE = 3.487. All other tests yielded non-significant results (as reported in detail in the S1 Data file, section **6.1.1.**).

## Discussion

Although the experience of being imitated is theorised to be a crucial driver of infant socio-cognitive development, empirical research on the topic remains scarce, especially when compared to the complementary ability of imitation production. In particular, the effects of being imitated and the ontogenetic course of imitation recognition have not been systematically investigated in early infancy.

In the present study, we attempted to tackle some of these issues, and found that 6-month old infants reliably detected that they were being imitated by an experimenter (as opposed to responded to contingently). They did so not only implicitly, by exhibiting increased levels of social signals (attention, smiling) and prosocial inclination when imitated, but also by exhibiting so-called 'testing behaviours', which are the hallmark of 'high-level' or 'explicit' imitation recognition [4,5,7,10,20]. All infants in the study but one exhibited testing behaviours, and the majority of testing bouts were accompanied by smiling. Behavioural repetition, which is regarded as the simplest form of testing behaviour, was present in nearly all testing bouts. The majority of these bouts, however, were mixed, comprising behavioural modulations and/or testing sequences as well. The presence of these more complex responses warrants the conclusion that the infants recognised that they were being imitated, and did not simply try to reproduce an interesting effect that their actions had on the experimenter, by repeating those actions.

Although testing behaviours are generally considered an indication that the imitated individual grasps the interlocutor's intentions to imitate [4,5,7,10], their presence in infancy has received a dual interpretation as a function of age [20]. More specifically, 9-month old infants, who are *a priori* regarded as unable to construe others as intentional agents, are said to make a

*behavioural* discrimination, driven by simple features, such as the detection of temporal contingency. In contrast, from 14 months of age, infants would make a *mental* discrimination, driven by the recognition of the experimenter's imitative intentions [20]. This interpretation is, however, problematic, and not supported by age-related differences in performance scores in tasks that probe socio-cognitive skills hypothesised to be related to imitation and intention recognition, such as gaze and point following, pretend play or mirror self-recognition [20,35,43]. Moreover, experimental studies on infant imitation recognition have consistently controlled for temporal contingency detection by including a CR condition. It is possible, however, that infants discriminate being imitated from other types of CR by relying on other kinds of low-level features, such as spatial compatibility or emotional signals. Unlike previous research on infant imitation, in this study we have included two novel conditions aimed at controlling for this possibility: contralateral imitation (CI) and bodily imitation without emotional mimicry (BI).

We found that implicit and explicit imitation recognition was not affected by the configural manipulation used in the CI condition, in which the left-right axis was inverted. This indicates that the infants' responses to being imitated (i.e. higher levels of attention, smiling, approach and testing behaviours) cannot be explained by simpler mechanisms, such as the detection of spatiotemporal contingency between their own and the experimenter's actions. Instead, in spite the left-right axis inversion, the infants recognised the structural similarity between their own and the experimenter's actions, i.e. they evidenced imitation recognition.

Generally, patterns of social responsiveness were consistent with an interpretation that social engagement was higher in the MI and CI conditions compared to the BI and CR conditions. Indeed, in the MI and CI conditions, attention engagement lasted longer (overall and average bout duration) and was more sustained than in the BI and CR condition, as there were fewer attention disengagements in the former than in the latter conditions. The higher frequency of attention bouts during the BI and CR conditions (compared to the MI and CI conditions) indicate that the infants continued to monitor the experimenter's behaviours, but probably found the experimenter's behaviour during these conditions less engaging. In addition, there were condition effects on smiling and approach that differentiated the MI and CI condition from the CR condition. As such, while smiling duration was affected, being longer in the MI and CI conditions compared to the CR condition, the frequency of smiling bouts was not. Similar effects were also found when comparing the MI with the BI condition, with smiling bouts being significantly shorter in the latter. This suggests that infants initiated smiling with equal frequency in the four conditions, but disengaged from such affective engagements much more quickly in the CR condition compared to the MI and CI conditions, and in the BI condition compared to the MI condition. Approach was affected with respect to both duration and frequency, as infants approached the experimenter more seldom and for shorter bouts during the CR condition compared to the MI and CI condition. The presence of bodily imitation in the BI condition appears to have attenuated such effects, as approach duration and frequency was not different in this condition compared to the MI and CI conditions.

Interestingly, while the suppression of emotional mimicry in the BI condition affected generic measures of social engagement (attention, smiling, approach), it did not affect measures of high-level imitation recognition (i.e. testing behaviours). This constitutes additional evidence consistent with the interpretation that the infants explicitly recognized that they were being imitated rather than discriminating imitation from CR based on low-level features. As such, there were no significant differences between the MI, CI and BI conditions with respect to the overall duration and frequency of testing behaviours between. Moreover, testing behaviours occurred significantly more often, lasted significantly longer, and were initiated with significantly shorter latencies in each of the imitation conditions compared to the CR condition.

However, latency to the first testing bout in the BI condition, while being significantly shorter than in the CR condition, was also significantly longer compared to the MI condition. We explain these effects as being most likely exerted on the *emotional* and *communicative* level rather than the *behavioural* one. In other words, the suppression of emotional mimicry did not disrupt infants' ability to recognise the behavioural correspondence between their own and the experimenter's actions. Instead, infants had difficulty grasping the interactional intentions of the experimenter, i.e. whether the experimenter would maintain a playful engagement in the interaction. Indeed, during the BI condition, infants were confronted with contradictory signals of engagement: the body of the experimenter was playful and mirrored the infant's bodily actions, while the experimenter's face remained immobile and silent.

Taken together, these results run against the dual interpretation previously proposed to account for the presence of testing behaviours in 9-month old infants as opposed to 14-month old infants or older [20]. While dualistic perspectives on mind and body construe intentions as 'mental plans' that are inaccessible to perception, assuming that their recognition in others requires detached, recursive representation, and an ability to make inferences, more recent theoretical approaches endorse a processual and embodied view to intention understanding. According to these approaches, intention awareness is emerging gradually, being contextually and experientially grounded in the infant's engagement with caregivers [44–46]. Indeed, from birth, infants are constantly experiencing the caregivers' intentional actions, initially as a target of them, and subsequently as participants in them [45,46]. At 6-months of age, one way in which infants engage with others is through rudimentary intentional imitation of others' actions [1,2,47,48]. Since first-person experience of a behaviour enhances the perception and recognition of that behaviour [49], it is fully conceivable that 6-month old infants explicitly recognise the imitative behaviours of others. In addition, since first-person experience of agency and intentionality facilitates the perception of others as intentional agents, and since 6-month old infants show rudimentary intention recognition [49], it is also plausible that they recognised the experimenter's imitative behaviour as intentional in our study. More specifically, considering the high frequency of infant smiles during testing bouts, the infants could have detected an intention to engage playfully.

This resonates well with the fact that testing behaviours, when resulting in a back-and-forth exchange of actions, may give rise to so-called 'imitation games', which are accompanied by smiling. In turn, the presence of such expressions of enjoyment and playfulness in response to being imitated has been interpreted as an indication of so-called 'shared intentionality' [50]—a suite of abilities that support the involvement in collaborative activities with joint goals and intentions [51]. This expression of shared intentionality appears thus to emerge early in human ontogenesis. In support of this interpretation, joint attention, which is regarded as the foundational element of shared intentionality, emerges at 6-months of age, if not earlier [46,52]. Moreover, emergent evidence also shows that imitation games—and social games in general—constitute an evolutionarily ancient form of shared intentionality that is present in all other great ape species [53,54].

It is also interesting to note that levels of social responsiveness in the BI condition were similar to those measured in the CR condition, in spite of the fact that the experimenter maintained a still-face during the BI condition. Indeed, from an early age, social contingency is the default interactional expectation of infants, as infants expect others to be responsive to their actions [3,7,10,13,21,22,24,32]. The still-face, in particular, has highly disruptive effects, with infants becoming distressed and eventually withdrawing from the interaction [39]. Given the robust evidence on both preference for contingent responding and still-face effects, it would have been possible that the similar levels of emotional engagement between the BI and CR conditions were the result of a deficient operationalisation of social contingency. We found

this not to be the case, as the levels of social responsiveness in the CR condition were similar to those measured during spontaneous mother-infant interaction in a matched control group. While the experimenter's responsiveness was higher in the CR condition (vocal, facial and gestural) compared to the BI condition (only gestural), these differences did not seem to lead to differences in the infants' level of social engagement. Indirectly, the patterns of responsiveness documented in this study suggest that, at 6-months of age, infants show higher levels of affective engagement when being imitated compared to regular mother-infant interaction. If so, this points to an important development in imitation recognition and imitation-induced prosociality between 3 and 6 months of age, since 3-month old infants exhibit less [14] or similar affective engagement [11,12] when being imitated compared to regular mother-infant interaction [13].

A potential limitation of the CR condition, which was raised by one of the reviewers, was that the experimenter's actions during the CR were all accompanied by vocalisations, and thus there was a risk that the experimenter responded to an infant vocalisation with a vocalisation, thus imitating the infant by chance. Rather than being a case of imitation, however, this is a case of channel congruence. While responding to an infant vocalisation with a vocalisation appears to be the most frequent form of maternal CR [22], to our knowledge this has never before been described as a form of imitation. It is unlikely that infants perceived vocal channel congruence as vocal imitation, considering that infants experience and detect vocal mimicry by caregivers from an early age [9,55], and can discriminate vowels at 3-months of age, thus being able to differentiate vocal matches from mismatches [56]. Moreover, it was very unlikely that the experimenter's vocalisations could match in *form* the vocalisations produced by the infants, since the experimenter's vocalisations, while potentially recognisable by the infants, were not part of the infant's vocal repertoire. It is worth pointing out that channel congruence is also present when the experimenter responds to an infant's manual gesture with a manual gesture that is temporally contingent but different from that produced by the infant, i.e. similarly to how the CR condition has been designed in two previous studies on imitation recognition with older infants.

It is also interesting to contrast the patterns of social responsiveness as measured in the experimental conditions to those measured in the SF intervals. As such, during the experimental intervals, social responsiveness was generally affected by condition, but not by presentation order. Conversely, during the SF intervals, social responsiveness was affected by order and, to a less extent, by sex. The lack of a condition effect on social responsiveness during the SF intervals is consistent with results from the only previous study that compared imitation and CR, and included SF-intervals between experimental conditions [20].

In the SF intervals there was a selective order effect which affected the duration—but not the frequency—of attention and smiling during the SF-intervals. As such, attention and smiling lasted significantly longer during the first SF interval compared to each of the three subsequent intervals. Moreover, social bidding was significantly more frequent in the first compared to the last SF interval. This shows that infants' attempts to re-engage in interaction with the experimenter were more insistent during the first SF interval, i.e. when the SF represented a novel type of situation. It is thus possible that the infants quickly grasped the temporal structuring of the experimental phase (i.e. with the experimenter alternating longer phases of activity with shorter phases of inactivity), as their attempts to re-engage became less insistent starting with the second SF interval. Similar order effects have been reported even in other studies that used repeated exposure to SF intervals [57]. Finally, the presence of sex effects during the SF-intervals is consistent with some previous studies that reported a higher propensity for girls to show higher levels of social responsiveness, although an alternative pattern has also been reported sometimes (see e.g. [58] for a review).

Taken together, the results of our study indicate that 6-month old infants discriminate between various levels of interactional contingency, and prefer the high levels of contingency that characterise imitative interactions. It has to be acknowledged, however, that these results are based on a relatively small sample of infants, that included predominantly male participants. Although power analyses indicated that, for the effects of condition reported in this study, minimal sample size requirements are met, additional studies should be conducted with a larger sample to both replicate and extend this research. Nevertheless, the present study has theoretical and methodological implications for research on social contingency and imitation recognition, calling for a refinement of definitions and analytical approaches in order to better highlight the interactional ingredients that are implicitly or explicitly detected and preferred at different ages during infancy. In addition, the study has implications for research on infants' perception of bodily communicative signals and their awareness of others' intentions toward them. We hope that our study will stimulate a more systematic investigation of imitation recognition in infancy, and a diversification of research toward understanding the mediating mechanisms of this ability, as well as its immediate and long-lasting consequences.

## Supporting information

**S1 Data.**
(DOCX)

**S1 Dataset.**
(XLSX)

**S2 Dataset.**
(XLSX)

**S3 Dataset.**
(XLSX)

**S1 Video.**
(MOV)

**S2 Video.**
(MOV)

**S3 Video.**
(MOV)

**S4 Video.**
(MOV)

## Acknowledgments

We heartily thank the families that signed up for the study. We are grateful for the reviewers' comments and observations.

## Author Contributions

**Conceptualization:** Gabriela-Alina Sauciuc, Tomas Persson, Sara Lenninger, Elainie Alenkaer Madsen.

**Data curation:** Gabriela-Alina Sauciuc, Jagoda Zlakowska.

**Formal analysis:** Gabriela-Alina Sauciuc.

**Funding acquisition:** Gabriela-Alina Sauciuc, Tomas Persson, Sara Lenninger, Elainie Alenkaer Madsen.

**Investigation:** Gabriela-Alina Sauciuc.

**Methodology:** Gabriela-Alina Sauciuc.

**Project administration:** Gabriela-Alina Sauciuc.

**Visualization:** Gabriela-Alina Sauciuc, Tomas Persson.

**Writing – original draft:** Gabriela-Alina Sauciuc.

**Writing – review & editing:** Tomas Persson.

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
