## [Decision Letter · Decision Letter 0]

3 Feb 2020

PONE-D-19-33798

Imitation recognition and its prosocial effects in 6-month old infants

PLOS ONE

Dear Dr. Sauciuc,

Thank you for submitting your manuscript to PLOS ONE. After careful consideration, we feel that it has merit but does not fully meet PLOS ONE’s publication criteria as it currently stands. Therefore, we invite you to submit a revised version of the manuscript that addresses the points raised during the review process.

As you can see one of the reviewer asks for some clarifications on the Method section. Methods are extremely important also because, if well explained, they permit the replication of the study. So I strongly suggest the authors to follow the recommendations of the reviewer. 

We would appreciate receiving your revised manuscript by April 1st, 2020. To enhance the reproducibility of your results, we recommend that if applicable you deposit your laboratory protocols in protocols.io, where a protocol can be assigned its own identifier (DOI) such that it can be cited independently in the future. For instructions see: http://journals.plos.org/plosone/s/submission-guidelines#loc-laboratory-protocols

We look forward to receiving your revised manuscript.

Kind regards,

Elisabetta Palagi

Academic Editor

PLOS ONE

Reviewers' comments:

Reviewer's Responses to Questions

**Comments to the Author**

1. Is the manuscript technically sound, and do the data support the conclusions?

Reviewer #1: Yes

Reviewer #2: Partly

2. Has the statistical analysis been performed appropriately and rigorously? 

Reviewer #1: Yes

Reviewer #2: Yes

3. Have the authors made all data underlying the findings in their manuscript fully available?

Reviewer #1: Yes

Reviewer #2: Yes

4. Is the manuscript presented in an intelligible fashion and written in standard English?

Reviewer #1: Yes

Reviewer #2: Yes

5. Review Comments to the Author

Reviewer #1: The manuscript is well written and it sound. It is easy to follow and I think it absolutely deserve to be pulished on Plos One.

Matherials and Methods are accurate and results are shown in a very accurate and clear way.

The only thing that is not so clear is the meaning of the asterisks in the figure. What does one asterisk means? and to asterisks? I think authors should explicit the meaning in the figure legend.

Reviewer #2: The manuscript by Sauciuc and colleagues aimed at investigating imitation recognition abilities in 6 month-old infants as well as whether being imitated affected their prosocial behaviours toward the experimenter. In particular, authors wanted to further explore whether young infants were able not only to show implicit awareness of being imitated but also explicit imitation recognition. Infants’ social responses were measured in 4 different experimental conditions: 1) mirroring imitation (MI) in which the experimenter matched infants’ actions on a temporal, spatial, morphological, topographical and affective level; 2) non-imitative contingent condition (CR) in which the experimenter performed one of 3 predetermined different actions every 20 seconds; 3) contralateral imitation (CI), similar to MI but infant’s actions were imitated contralaterally; 4) body imitation condition (BI) where only body movements were imitated ipsilaterally, while all the facial and vocal feedback was suppressed. While the first two conditions have been frequently used in previous studies on infant imitation recognition, the CI and BI conditions were added by the authors to control for spatial compatibility effects and the contribution of emotional mimicry to the social interaction respectively.

In agreement with previous studies, authors found that, when being imitated, infants show higher level of prosocial behaviours. Interestingly, they also found higher rates of testing behaviours in all the imitative conditions, suggesting evidence of both implicit and explicit imitation recognition in 6 month-old infants. Moreover, the suppression of the emotional mimicry in the BI condition lead to a decrease in infant prosocial behaviours, without affecting the high-level imitation recognition skills, suggesting an early ability to infer intentionality of imitative actions.

This is potentially a very interesting and informative study on infant imitation recognition, as it could expand our understanding of the communicative value of imitative behaviours in infancy, as well as provide important information about the different mechanisms involved in the development of implicit vs explicit imitation recognition. However, this manuscript in its current version has significant methodological flaws and requires major revisions in order to meet the journal’s requirements.

The authors should provide a more detailed explanation of the following:

1. Methods

CR condition - Despite the detailed and comprehensive evaluation of the control conditions used in previous studies on infant imitation recognition and the attempt by the authors to overcome any potential limitations by using a different approach, I have some concern about the CR condition used in the current study. The purpose of a non-imitative contingent control condition in imitation studies is to disentangle the effect of temporal contingency from the one driven by the structural (and temporal) contingency. In fact, in both human and non-human primates, it has been shown that subjects recognize when being imitated only when the action performed by the imitator is both temporally AND structurally contingent as well as attuned in terms of affective valence. In the current study, the authors decided to use a pre-set inventory of 3 actions and presented them every 20 seconds. Thus, the experimenter performed a total of 6 gestures in 2 minutes, which I assume is far less than the number of actions (i.e. body movements, facial expressions, vocalizations) an infant would perform in 2 minutes. This poses several issues. First, the infant did not received a contingent response any time she performed an action, with a level of contingency experienced lower than in the other conditions. Second, by using one of the 3 predetermined responses, the infant received a random response to her actions, which could be contingent not only temporally, but also structurally and/or attuned in terms of affective valence, therefore resembling an imitative interaction more than a non-imitative contingent one. For example, if the infant waved her hand and by chance the imitator used as a response the ‘waving while saying Hello’ response, then the infant would experiences a temporal and structural contingency. In addition, the presence of the vocalization would increase the salience of the response as well as its emotional content, thus resulting in a response very different from a non-imitative contingent response. Finally, the addition of a vocalization in all the responses provided by the imitator poses another methodological issue. In fact, whenever the infant performed an action in one modality only (e.g. a vocalization, a mouth gesture, a hand gesture), she would always receive a response in two different modalities, increasing the salience of the response and making the interaction more resembling a natural social interaction (as shown also by the results of the comparison between CR and the naturalistic mother-infant interactions).

If the scope of having a control condition is to show that temporal contingency alone is not effective in allowing infants to recognize they are being imitated and increasing their prosocial behaviours, then the variables mentioned above should be controlled more rigorously.

Furthermore, to assess the efficacy of the CR condition, authors compared the levels of infant attention and smiling measured in this condition to those measured during a naturalistic mother-infant interaction. The two conditions are not comparable on any level, and it is not clear why the authors decided to make this comparison. If the scope was to see whether the interaction between the experimenter and the infant was affected by the non-familiarity with the imitator, then a comparison should be done between the interaction between mothers and infant and the preparation phase described on page 9.

Alternatively, the authors could have asked the mothers to interact with their babies according to the same procedure used in the CR condition, thus making the two observations comparable.

CI condition – this condition slightly differs from the MI one, as only the hand and trunk movements are imitated contralaterally, while all the facial gestures and vocalizations are imitated exactly as in the MI condition. The similarity between the two conditions might explain why no difference has been found in any of the infant behaviours between the MI and the CI conditions.

Still face interval – the duration of the still-face phase should be similar to the one used for the other experimental conditions (i.e. 2 minutes). Even if the length of the still-face phase has varied between 30 and 180 seconds in previous studies, for a more rigorous experimental design and in order to investigate the still-face effect on infant behaviours, the same duration should be used for the interactive phase and the still-face one.

Moreover, similarly to previous studies, the authors should include non-distress vocalizations among the behaviours scored during the still-face phase, as it have been shown that infant increase the frequency of such behaviours after an imitative interaction in the attempt to re-engage the interaction with the imitator.

2. STATISTICAL ANALYSIS

The authors should have coded infants’ social and testing behaviours in both the interactive phase of the interaction and the still-face one, keeping the two phases separated. From the analysis it is not clear whether the behaviours scored in the 30 sec of still face were not included in the analysis or were scored together with the first 2 minutes of interaction.

An analysis of the behaviours shown by the infants during the still-face period would allow authors to better understand whether infants are sensitive to the quality of the interaction therefore showing more prosocial behaviours toward the imitator as well as whether they perceive themselves as active agents within the interactions by showing more social bids and testing behaviours.

3. RESULTS AND DISCUSSION

Overall, the authors should discuss each of the results found more extensively and with more support.

In particular, for all the infant behaviours included in the analysis they should further discuss the similarity between the MI condition and the CI one. As stated before, the two conditions differ very slightly to each other, and this could explain why no difference was found. Alternatively, results should be discussed in term of structural resemblance in both conditions despite the inversion in the left-right axis.

I agree with the decision of scoring infant behaviours in term of duration, frequency, and mean duration of each bout, as different measures can indeed provide different information about the interaction. However, authors did not discuss their findings in this regard. For example, they found that in MI and CI conditions infant looked more to the experimenter, less frequently but longer in each bout, and discuss this in terms of implicit recognition and more social engagement during the imitative interactions with high level of contingency. However, they did not discuss that infants in BI and CR, despite showing less social engagement, looked more frequently at the experimenter, suggesting evidence for monitoring behaviours towards him/her.

Similarly, infants smiled longer in the MI and CI conditions, but the frequency of smiles was not different among the four different conditions, suggesting again an effect of imitation on social engagement but also a possible use of social signals in trying to engage socially with the experimenter in the other two conditions. In this regard, analyzing the data according to the experimental phase (interaction vs still face) would give more information about the quality of the interaction and the active role of the infant in modulating the interaction.

Another interesting result that would be worth discussing further is that infants showed the same amount and duration of testing behaviours in the MI, CI and BI conditions. Interestingly, the suppression of emotional mimicry lead to a decrease in social engagement, but did not prevent infant to detect behavioural similarities between their own actions and those performed by the experimenter and show testing behaviours toward him.

Again, it would be interesting to see whether the same results would be obtained when scoring testing behaviours during the two phases of the interaction.

Minor issues:

The introduction would benefit from the inclusion of more previous literature on imitation recognition and maternal mirroring in young infants and non-human primates (e.g. Beebe et al., 2003; Meltzoff, 2007; Murray et al., 2014; Murray et al., 2016; Sclafani et al., 2014).

Table 4 – Check the frequency of attention in the CR experimental condition and the control mother-infant interaction. The value reported are 12.438 (1.021) and 1.355 (12.065) respectively. I assume the second value is not correct, as no difference was found between the two conditions, and considering the extremely high SEM value.

Authors should explain why they used the Benjamini-Hochberg correction for multiple comparisons instead of the Bonferroni one.

6. PLOS authors have the option to publish the peer review history of their article (what does this mean?). If published, this will include your full peer review and any attached files.

Reviewer #1: No

Reviewer #2: No

---

## [Author Response · Author response to Decision Letter 0]

31 Mar 2020

Response to reviewers has been appended as a separate file

---

## [Decision Letter · Decision Letter 1]

21 Apr 2020

Imitation recognition and its prosocial effects in 6-month old infants

PONE-D-19-33798R1

Dear Dr. Sauciuc,

We are pleased to inform you that your manuscript has been judged scientifically suitable for publication and will be formally accepted for publication once it complies with all outstanding technical requirements.

With kind regards,

Elisabetta Palagi

Academic Editor

PLOS ONE

Reviewer's Responses to Questions

**Comments to the Author**

1. If the authors have adequately addressed your comments raised in a previous round of review and you feel that this manuscript is now acceptable for publication, you may indicate that here to bypass the “Comments to the Author” section, enter your conflict of interest statement in the “Confidential to Editor” section, and submit your "Accept" recommendation.

Reviewer #1: All comments have been addressed

Reviewer #2: All comments have been addressed

2. Is the manuscript technically sound, and do the data support the conclusions?

Reviewer #1: Yes

Reviewer #2: Yes

3. Has the statistical analysis been performed appropriately and rigorously? 

Reviewer #1: Yes

Reviewer #2: Yes

4. Have the authors made all data underlying the findings in their manuscript fully available?

Reviewer #1: Yes

Reviewer #2: Yes

5. Is the manuscript presented in an intelligible fashion and written in standard English?

Reviewer #1: Yes

Reviewer #2: Yes

6. Review Comments to the Author

Reviewer #1: (No Response)

Reviewer #2: All comments have been addressed, and the manuscript is very much improved compared to the first submission.

7. PLOS authors have the option to publish the peer review history of their article (what does this mean?). If published, this will include your full peer review and any attached files.

Reviewer #1: No

Reviewer #2: No

---

## [Editor Report · Acceptance letter]

27 Apr 2020

PONE-D-19-33798R1 

Imitation recognition and its prosocial effects in 6-month old infants 

Dear Dr. Sauciuc:

I am pleased to inform you that your manuscript has been deemed suitable for publication in PLOS ONE. Congratulations! Your manuscript is now with our production department. 

With kind regards,

on behalf of

Dr. Elisabetta Palagi 

Academic Editor

PLOS ONE